# The Dynamics of Hole Transfer in DNA

**DOI:** 10.3390/molecules24224044

**Published:** 2019-11-07

**Authors:** Andrea Peluso, Tonino Caruso, Alessandro Landi, Amedeo Capobianco

**Affiliations:** Dipartimento di Chimica e Biologia “A. Zambelli”, Università di Salerno, via Giovanni Paolo II, 132, I-84084 Fisciano (SA), Italy; apeluso@unisa.it (A.P.); tcaruso@unisa.it (T.C.); alelandi1@unisa.it (A.L.)

**Keywords:** DNA oxidation, DNA hole transfer, DNA, quantum dynamics, electron transfer, charge transfer

## Abstract

High-energy radiation and oxidizing agents can ionize DNA. One electron oxidation gives rise to a radical cation whose charge (hole) can migrate through DNA covering several hundreds of Å, eventually leading to irreversible oxidative damage and consequent disease. Understanding the thermodynamic, kinetic and chemical aspects of the hole transport in DNA is important not only for its biological consequences, but also for assessing the properties of DNA in redox sensing or labeling. Furthermore, due to hole migration, DNA could potentially play an important role in nanoelectronics, by acting as both a template and active component. Herein, we review our work on the dynamics of hole transfer in DNA carried out in the last decade. After retrieving the thermodynamic parameters needed to address the dynamics of hole transfer by voltammetric and spectroscopic experiments and quantum chemical computations, we develop a theoretical methodology which allows for a faithful interpretation of the kinetics of the hole transport in DNA and is also capable of taking into account sequence-specific effects.

## 1. Introduction

Eley and Spivey first envisioned DNA as a possible conduit for conveying electrical charges, via the π system of stacked nucleobases [1]. However, long-range charge transport in DNA was discovered only in the 1990s, by Barton and coworkers [2].

Since then, a large body of experimental evidence has accumulated, showing that one electron oxidation on a DNA donor site (D) produces a hole that can migrate through the double helix, covering long distances (up to several hundreds of Å) until an irreversible oxidative damage takes place at an acceptor site (A) (see Figure 1) [3,4,5].

Living cells are continuously exposed to endogenously generated as well as external agents that can oxidize DNA [6]. That may result in the corruption of genetic information with potentially serious consequences, including mutagenesis and cancer [7].

Aside from its enormous biologic relevance, long-range hole transport in DNA has attracted much interest because: (*i*) it is useful for detecting structural changes in DNA resulting from alterations of the regular π–π stacking [8,9]; and (*ii*) it enables a potential use of DNA as a dielectric material in field-effect transistors and organic light-emitting diodes, hopefully leading to biosustainable devices [10,11,12,13,14,15].

The stability and the conformational flexibility of double stranded DNA largely result from the interaction of water and counterions with the charged sugar-phosphate backbone [16]. Dielectric effects strongly affect the oxidation of nucleobases inside DNA, even if the nucleobases experience a strongly hydrophobic environment as opposite to the charged phosphate backbone. Indeed, the oxidation potentials of oligonucleotides and short B-DNA sequences inferred from voltammetric measurements were found to be strongly dependent on pH (due to possible pH mediated proton transfer) and counterion concentration [17,18,19,20,21,22,23]. Moreover, molecular dynamics simulations and quantum chemical computations revealed that structural fluctuations causing the redistribution of Na${}^{+}$ counterions and their associated water molecules strongly affect the energy of the HOMO levels of the nucleobase units inside DNA [24].

Aside from B-DNA, a very efficient hole transport has been observed also in guanine quadruplex stacked sequences adsorbed on a mica substrates [25]. Indeed, in recent years, growing attention has been given to the oxidation of G-quadruplexes, for they occupy telomeric regions of chromosomes, often found in oncogene promoter sequences [26].

HT is known to be strongly dependent also on the specific sequence of nucleosides and on DNA conformation. In view of all previous considerations, it is clear that addressing the dynamics of hole transport (HT) from a theoretical viewpoint constitutes a very difficult task. Several approaches have been used to study the kinetics of hole transfer in double stranded B-DNA and different conclusions about its underlying mechanism have been reached so far [27,28,29,30,31,32,33,34,35,36,37,38,39,40]. Nevertheless, a few firm points have been established. The observed products of the one-electron oxidation are rather insensitive to the process by which DNA is oxidized [4]. For DNA sequences containing guanine (G), the final radical cation usually localizes at G sites [6,7,41], because guanine is the most readily oxidized natural occurring nucleobase. Runs of two or more adjacent Gs are often used as thermodynamic traps to localize the hole, which is then detected as an alkali-labile lesion, because DNA steps composed of consecutive guanines experience a further lowering of the oxidation potential [42,43,44,45,46,47]. Furthermore, it is now well assessed that steps composed of adjacent adenine (A) nucleobases greatly facilitate the hole transport [18,21,48,49,50], while consecutive stacked thymines (T) and cytosines (C) act as barrier sites, strongly attenuating hole transfer efficiency, due to their higher ionization energy [4].

Herein, we review the work carried out by our research group in the last decade on the dynamics of hole transfer in DNA, focusing particularly on electrochemical measurements and showing how their outcomes have led to building up a quite general kinetics model for treating hole transfer in DNA and other molecular wires.

## 2. Hole Site Energy

The dynamics of HT in DNA is modulated by two quantities: (*i*) the hole energies of the nucleobases, the actual redox sites in DNA [51]; and (*ii*) the electronic couplings between adjacent sites. Both the observed oxidation free energies of nucleobases, nucleosides and oligonucleotides in aqueous environment and the hole trapping efficiencies of DNA sequences originate from those quantities.

Although the importance of dielectric effects has been fully recognized, until recently, the majority of studies concerning long range hole transfer in DNA employed hole site energies inferred from the gas phase [29,32,33,34,52,53,54,55]. That is far from being satisfactory because the actual ionization energy of a nucleobase in hydrated DNA is strongly affected by dielectric effects, hydrogen bonding of complementary bases, and, above all, intrastrand π–π stacking interactions [18,56,57].

The redox properties of nucleobases, nucleosides, (oligo)nucleotides and DNA sequences have been deeply investigated by voltammetric techniques [17,58,59]. Although cyclic voltammetry measurements show the presence of collateral reactions leading to irreversible processes [60,61], the hole site energy spacing inferred from electrochemistry closely matches the one obtained by liquid-jet photoelectron spectroscopy (PES) not suffering from the above problem [56].

A selection of hole site energies of DNA constituents obtained by different techniques is reported in Table 1.

With the exception of spectroscopic measurements (sixth column), whose reliability for pyrimidine derivatives is quite modest [64,65], a substantially good agreement is found for the data of Table 1 referring to solvated environment. PES, voltammetry, and quantum chemical predictions find guanine as the most easily oxidizable nucleobase; the hole energy of adenine is ≈0.4 eV higher than guanine, while pyrimidine derivatives are oxidized at a potential higher by 0.6–0.8 eV than G.

A graphical comparison of the hole energies for the aqueous environment (first column of Table 1) with those referred to gas phase (last column of Table 1) is presented in Figure 2.

Solvation acts by somewhat leveling the hole energies of DNA constituents. While the hole energy of adenine (relative to guanine) is scarcely affected by solvation, cytosine, and, above all, thymine become comparably easier to ionize in solution. Indeed, oxidative damages are often observed at T sites in oligonucleotides which lack G [67,68,69,70].

The data in Table 1 do not include the effects of the H-bonded complementary base on ionization energies of nucleobases. The lowering of the oxidation potential of G due to the base pairing with C had been predicted by theoretical computations [71], and experimentally estimated by the increase of the oxidation rate of guanosine (Guo) upon cytidine (Cyd) pairing [57,72]. However, a direct measurement of the above quantity was not available until 2005, when an electrochemical study carried out in our laboratories settled the question [73].

Guanosine and deoxycytidine (dCyd) were properly functionalized to make them soluble in chloroform, a solvent in which the association constant for the formation of the Watson–Crick Guo:dCyd H-bonded complex is sufficiently high to permit its detection [74,75]. Then, voltammetric measurements of solutions containing Guo, dCyd, and their mixtures were carried out. The same procedure was later adopted for the adenosine (Ado) deoxythymidine (dThd) pair [76]. The main results of our investigations are summarized in Figure 3. The differential pulse voltammogram of the solution containing an equimolar amount of Guo and dCyd (Figure 3a) shows two well-resolved peaks, one occurring at the same potential observed for solutions containing only the Guo nucleoside, which can therefore be assigned to the fraction of free Guo in solution, and the other occurring at a potential lower by 0.34 V is assigned to the Guo:dThd Watson–Crick complex. Ado:dThd voltammograms exhibit a similar behavior and identical conclusions were inferred for the Ado:dThd hydrogen bond complex (Figure 3b).

H-bond association with complementary nucleosides lowers the oxidation free energy of Guo and Ado by ca. 0.3 eV, because ionization causes a substantial increase of the binding energy in the oxidized Watson–Crick complex with respect to its neutral counterpart [77].

The voltammograms in Figure 3 show no anodic signal for pyrimidine nucleosides in chloroform. Indeed, oxidizing pyrimidine derivatives in solution is a very difficult experimental task [58,64,78]. Nevertheless, the first estimate of the energy of a low lying excited state of DNA with the hole localized on cytidine in the Watson–Crick complex with guanosine was inferred by spectroelectrochemistry measurements [79].

NIR spectra of solutions containing Guo:dCyd mixtures were recorded in an electrochemical cell equipped with an optically transparent thin-layer electrode kept at +0.57 V versus Fc${}^{+}$/Fc in CHCl${}_{3}$ and CH${}_{2}$Cl${}_{2}$. At that potential, solutions containing only Guo or only dCyd are not oxidized, whereas solutions containing both species exhibit a well-resolved anodic peak (Figure 3). A positive broad band (Figure 4), not observed during the oxidation of solutions containing only Guo or only dCyd, was recorded in the difference spectrum, at approximately 10,600 cm${}^{-1}$ in CH${}_{2}$Cl${}_{2}$ and at 10,200 cm${}^{-1}$ in CHCl${}_{3}$. Upon replacing cytidine with 5-methylcytidine, whose ionization energy is expected to be lower than that of cytidine by ca. 1400 cm${}^{-1}$ [66,79], that band was red shifted to 9100 cm${}^{-1}$ in CH${}_{2}$Cl${}_{2}$ and to 8700 cm${}^{-1}$ in CHCl${}_{3}$. On the basis of the above evidence and with the support of time dependent DFT (TDDFT) computations, that signal was assigned to the charge-transfer (CT) localizing the hole on cytidine (Figure 4).

## 3. Electronic Couplings

Stacking interactions are by far the most important inter-base interactions for hole transfer because they provide the electronic couplings for long-range hole transfer. The effect of stacking interactions can be addressed by a simple two state quantum model, according to which the hole energy levels of two stacked nucleobases (X and Y) are shifted up and down with respect to those of unstacked ones by a quantity related to the difference between the hole energies of the two nucleobases $\varepsilon_{X}$, $\varepsilon_{Y}$, and to the strength of the stacking interactions, $J_{\mathrm{XY}}$.

In the case of identical nucleobases X=Y (Figure 5, left), the hole energy shift upon pairing of nucleobases is just the electronic coupling element. Instead (Figure 5, right), if $\varepsilon_{X}\gg \varepsilon_{Y}$, and $\varepsilon_{X}-\varepsilon_{Y}\gg 2J_{\mathrm{XY}}$, then the $J_{\mathrm{XY}}$ coupling term is not effective in lowering the hole energy of the stack, so that the lowest eigenvalue of the two state model Hamiltonian, H, is nearly coincident with $\varepsilon_{Y}$.

Provided that two identical Y nucleobases assume a regular conformation, i.e. they are efficiently stacked, the $J_{\mathrm{YY}}$ coupling term can be estimated as the lowering of the ionization energy of the YY stacked sequence, with respect to that of a strand containing only one Y oxidizable nucleobases, on the assumption that all other nucleobases have higher hole site energies.

The reliability of the two state model has been verified by voltammetric experiments [47,80]. Figure 6 reports the differential pulse voltammograms recorded in water for the 5${}^{\prime }$-ACCCCA-3${}^{\prime }$ and 5${}^{\prime }$-AACCAA-3${}^{\prime }$ single stranded DNA oligonucleotides. A lowering of the oxidation potential amounting to 0.31 V is observed for the sequence containing two consecutive adenines. Measurements carried out for sequences containing an increasing number of adjacent adenines end capped by thymine nucleobases confirmed that result; anodic peaks for the first oxidation were detected at 0.97, 0.90 and 0.82 V vs. Ag/AgCl for 5${}^{\prime }$-TTAATT-$3^{\prime }$, 5${}^{\prime }$-TTAAAT-$3^{\prime }$, and 5${}^{\prime }$-TAAAAT-$3^{\prime }$, single strands, respectively [80].

If oligonucleotides possessing adjacent adenines assume conformations in which nucleobases are well stacked altogether, as is indeed the case for A-rich tracts [80,81,82,83,84,85,86], which are known to confer structural rigidity to DNA [87,88], the two state model holds. Therefore, disregarding the coupling between A and C as a first approximation, $J_{\mathrm{AA}}$ was estimated to amount to ≈0.3 eV by the voltammograms of Figure 6 [80].

The observed progressive lowering of the oxidation potential upon increasing the number of consecutive stacked adenines is well predicted by PCM/DFT calculations carried out for the same single stranded DNA sequences used in experiments. Indeed, the computed ionization potential shifts are in very good agreement with the observed oxidation potentials (see Figure 7). Furthermore, the analysis of spin distributions (Figure 7) indicates that the observed oxidation potential shifts can be assigned to orbital mixing effects among stacked nucleobases, which lead to the formation of delocalized polarons [85]. Notably, there has been a vivid debate about the role of delocalized polarons in the charge transport in DNA [18,21,39,89,90,91,92,93,94,95,96,97,98,99,100,101]. Aside from voltammetric evidence [80], the formation of the $A_{n}^{•+}$ polaron has been later on observed also by time dependent spectroscopy measurements carried out for oxidized DNA hairpins possessing two or more intervening A-T steps [50].

A progressing lowering of the oxidation potential was also observed in single and double stranded oligonucleotides possessing an increasing number of consecutive guanines, showing that hole site energies can be lowered up to ≈0.3 eV for sites composed of six consecutive guanines [47]. The observed shift of the oxidation potential leads to $J_{\mathrm{GG}}\approx$0.1 V, an electron coupling significantly lower than that observed for adenine. That result stems from the lower extent of hole delocalization on G steps. Theoretical studies have concluded that the positive charge is almost entirely localized on ionized 5${}^{\prime }$-G in GG steps, possibly due to strong heteroatom-π electrostatic interactions [33,44,100,102,103,104,105,106,107,108]. Indeed, for DNA sequences containing up to two consecutive guanines, the oxidative damage is not equally distributed over G sites, but preferentially occurs at the 5${}^{\prime }$ G of the GG step [42,102,109]. However, DNA cleavage efficiency does not depend only on hole trapping efficiencies, but it also relies on the kinetic mechanism by which DNA damage occurs, therefore no unambiguous conclusion can be drawn for the preferred cleavage site based only on the preferred ionization site [67,68,106,110].

The extension of the set of electronic couplings to pyrimidine nucleobases is an experimentally very challenging task because of the well-known difficulties of oxidizing pyrimidine derivatives in solution, especially by voltammetric techniques [73,76,111]. Therefore, to retrieve the whole set of coupling parameters also including pyrimidine derivatives, we resorted to PCM/DFT computations, which for purine bases had proved to provide faithful descriptions of the available experimental data [85].

To evaluate the effects of π–π stacking on ionization energies separately from hydrogen bonding, we had to resort to single stranded DNA sequences. In detail, we considered the two sets of tetrameric single stranded sequences 5${}^{\prime }$-XXYX-3${}^{\prime }$ and 5${}^{\prime }$-XYZX-3${}^{\prime }$, in which Y and Z are natural occurring DNA nucleobases and X is a nucleoside analogue possessing an ionization energy much higher than DNA nucleobases. Single stranded tetramers are the simplest sequences in which both nucleobases of the YZ tract occupy an internal position inside the strand. That permits minimizing inconsistencies arising from different exposure to the solvent. Figure 5 (right) shows that any coupling of X with DNA nucleobases is ineffective on the oxidation potential of nucleobases. In that case, according to the two state model, the in situ hole site energy of a DNA nucleobase nearly coincides with the ionization energy of 5${}^{\prime }$-XXYX-3${}^{\prime }$, whereas $J_{\mathrm{YZ}}$ intrastrand coupling terms are given by:(1)$$J_{\mathrm{YZ}}=\sqrt{\Delta I\left(\Delta I-\Delta \varepsilon \right)},$$
where: (2)$$\begin{array}{cc} \Delta \varepsilon =\varepsilon_{Y}-\varepsilon_{Z} & \approx I_{\mathrm{XXYX}}-I_{\mathrm{XXZX}}, \end{array}$$(3)$$\begin{array}{cc} \Delta I=E_{\mathrm{YZ}}^{-}-\varepsilon_{Z} & \approx I_{\mathrm{XYZX}}-I_{\mathrm{XXZX}}, \end{array}$$
in which *I* denotes ionization potential. The assumptions underlying Equations (Equation 1)–(Equation 3) are illustrated in Figure 8 [62].

Hole site energies and coupling terms inferred by Equations (Equation 1)–(Equation 3), with X=6-azauracil (a nucleobase analogue with very high ionization energy due to the electron withdrawing effect of nitrogen [112], employed as a growth inhibitor of microorganisms which is known to incorporate into nucleic acids [113,114,115]), are reported in Table 2.

Hole transport properties, oxidation potentials, trapping efficiencies and several other properties of oxidized DNA can be addressed by the tight binding (TB) Hamiltonian commonly used for the dynamics of charge transport [34,54,116,117,118,119,120,121,122,123]:(4)$$H=\varepsilon_{L}|L\rangle \langle L|+\sum_{n=1}^{L-1}\varepsilon_{n}|n\rangle \langle n|+\left(J_{n,n+1}|n\rangle \langle n+1|+H.c.\right).$$

In H, only the interactions between nearest neighbor sites are considered. *L* is the number of nucleobases, $|n\rangle$ constitutes a set of orthogonal diabatic states with the charge fully localized on the *n*th nucleobase, $\varepsilon_{n}$ is the hole energy of $|n\rangle$ assumed to be independent of nucleobase sequence, and $J_{n,m}$ are the electronic coupling elements between $|n\rangle$ and $|m\rangle$, i.e., the interaction energies due to π–π stacking. Hole site energies and electronic couplings are the parameters to be employed in the model Hamiltonian.

In Figure 9, the ionization energies of several tetrameric single stranded DNA sequences predicted by DFT computations are compared with those obtained by diagonalizing the TB Hamiltonian of Equation (Equation 4) using the parameters inferred by Equations (Equation 1)–(Equation 3), (Table 2). Predictions by the TB Hamiltonian are in excellent agreement with the outcomes of DFT computations and also with experimental evidence. Ionization energies of TTAAAT and TTAATT single strands are found by TB to be higher by +0.05 and +0.13 eV, respectively, than that of TAAAAT, to be compared with the corresponding oxidation potentials inferred by voltammetric measurements: +0.08 and +0.15 V (see Figure 7).

## 4. The Dynamics of Hole Transfer in DNA

The kinetics of HT in DNA is known to exhibit two regimes: for shorter distance between the hole donor and the hole acceptor nucleobases the rate exponentially depends on distance, whereas a much weaker distance dependence is observed for longer donor–acceptor distances [39,45]. That behavior is exemplified by Giese experiments on double-stranded 3${}^{\prime }$-G(T)${}_{n}$GGG-5${}^{\prime }$ oligonucleotides, in which a hole is injected onto the single G site via photoexcitation of a suitably modified nucleotide, and the yields of oxidation products formed at the initial site ($P_{G}$) and at the trap site ($P_{\mathrm{GGG}}$) are measured [45]. Giese and coworkers observed that for shorter sequences, up to n=3, the product ratio $P_{\mathrm{GGG}}/P_{G}$ drops by a factor of ca. 8 for each additional adenine:thymine (A:T) step. In longer sequences, for n=4–7, the $P_{\mathrm{GGG}}/P_{G}$ ratio exhibits a much weaker distance dependence, whereas, for n=7–16, no substantial change in $P_{\mathrm{GGG}}/P_{G}$ was detected. Experimental results were interpreted by admitting a switch of HT mechanism from coherent superexchange in the short range regime to a thermally induced multistep or multirange hopping for the long range regime [28,36,39,122,126,127]. Nevertheless, theoretical models aimed at describing both the long- and short-range regimes in the framework of a single mechanism have also been proposed [122,128].

It is worth noting that the weak dependence of HT in the long range regime is consistent with different underlying mechanisms based on very different underlying physics [129]. Therefore, the detection of the mechanism for HT in DNA cannot rely solely on experimental observations. Theoretical modeling and simulations thus appear to be essential tools for solving such a high complex problem.

Following to some extent the recent work of Parson [130,131,132,133], we recently presented a novel methodology especially suited for addressing the dynamics of charge transport in molecular wires [134], which has been applied to the DNA oligomers investigated by Giese, whose work is particularly appealing, for it provides the yield ratios of the water trapping products of hole transfer for a large number of DNA oligomers [45].

Our approach relies on the multi-step kinetic model illustrated in Figure 10, in which D${}^{+}$(Bridge)A and D(Bridge)A${}^{+}$ denote the initial state with the charge localized on the donor and the final CT state, respectively; [D${}^{+}$(Bridge)A]* and [D(Bridge)A${}^{+}$]* denote the ensembles of structures in which the hole donor and acceptor are in vibronic resonance with each other; and $P_{D}$ and $P_{A}$ denote the products of oxidative damage occurring at donor and acceptor sites, respectively. The HT mechanism starts with an activation step, which brings the donor and the acceptor groups into electronic degeneracy (Step 1); Step 2 represents the elementary electron transfer between resonant donor and acceptor groups, followed by relaxation of all the non equilibrium species to their minimum energy structures (including solvent) and formation of hole transfer products (Step 3).

Step 3 and the reverse of Step 1 take into account the solvent response to a nonequilibrium charge distribution of the solute; pump–probe experiments in water solutions showed that solvent relaxation occurs in a few tens of femtoseconds, thus we set $k_{21}=k_{\mathrm{irr}}=10^{13}$ s${}^{-1}$ [135,136]. $k_{P}$ has been set to $10^{7}$ s${}^{-1}$, taken from the rate of deprotonation of Guo${}^{\cdot +}$ [137,138], likely the first step of formation of the products of oxidative damage [139,140,141].

Because the DNA oligomers studied by Giese contain several consecutive rigid A:T steps, it is possible to assume that Step 1 is governed only by solvent motion rather than by backbone reorganization. However, no experimental information about $k_{12}$ is available, therefore $k_{12}$ has been taken as an adjustable parameter to be inferred from experimental results.

Step 2 is the hole transfer, which is mainly governed by the nuclear motion of nucleobases [142,143]. Because of the local rigidity of the DNA backbone due to the (A:T)${}_{n}$ tract [88], we can make the reasonable assumption that the elements of the ensemble of the activated HT reactants differ from each other only for solvent configurations, which are irrelevant for calculation of the rates of the elementary HT step. Therefore, quantum dynamics computations can be carried out for only one of the typical equilibrium configurations of the different oligonucleotides. In our approach, the kinetic constant for the charge transfer step has been determined as $k_{23}=1/\tau_{23}$, where $\tau_{23}$ are transition times taken at the complete population of the final state, i.e., when the hole is fully localized on the GGG site. To compute transition times, we numerically solved the time dependent Schrödinger equation:(5)$$iћ\frac{\partial |\psi (t)\rangle }{\partial t}=H|\psi (t)\rangle .$$

Upon introducing the vibronic nature of the diabatic states in Equation (Equation 4), $|n\rangle \to |n\rangle ⊗|\nu \rangle \equiv |n,\nu \rangle$, where $|\nu \rangle$ denotes the manifold of the harmonic vibrational states of the *n*th nucleobase, the TB Hamiltonian can be cast in the form:(6)$$H=\sum_{n,\nu }\left(\varepsilon_{n}+E_{\nu }\right)|n,\nu \rangle \langle n,\nu |+\;\sum_{n,m,\nu ,\mu }J_{nm}\langle \mu |\nu \rangle |n\rangle \langle m|+\;\sum_{n,m,\nu ,\mu ,i}\frac{\partial J_{nm}}{\partial Q_{i}}\langle \mu \left|Q_{i}\right|\nu \rangle |n\rangle \langle m|,$$
in which the Born–Oppenheimer approximation has been used. $E_{\nu }$ denote vibrational energies, and the last summation has been introduced to take into account the possible fluctuations of the electronic couplings $J_{nm}$ due to interbase oscillations along the *i*th normal coordinate, causing dynamical deformation of regular double stranded DNA [52,91,144,145].

The $\partial J_{nm}/\partial Q_{i}$ factor has been kept fixed at 0.1 eV/Å, ε and *J* parameters have been taken from Table 2, and the G/A inter-strand coupling term has been set to 0.012 eV. Consistent with the Hamiltonian of Equation (Equation 6), the time dependent wave function is expanded over the set of Born–Oppenheimer products of time independent basis functions:(7)$$|\psi (t)\rangle =\sum_{n,\nu }C_{n,\nu }\left(t\right)|n,\nu \rangle .$$

To make calculations possible, the computational load connected with the huge number of integrals has to be strongly reduced. That goal is achieved by partitioning the Hilbert space into a set of subspaces with a fixed number of vibrations that are allowed to be simultaneously excited. Only the normal modes that are effectively coupled to hole transfer vibrations, i.e., the modes which are allowed to change their quantum number during the transition, are included in computations. Active modes are selected according to Duschinsky’s transformation, as the ones giving rise to the largest displacement of geometrical coordinates upon electronic excitation [121,134].

We have considered the possibility that in Giese sequences HT may occur either intrastrand, via T units, or interstrand, mediated by the adenine nucleobases of the complementary helix. The results of quantum dynamics simulations are reported in Figure 11, where the logarithm plot of the predicted kinetic constants of Step 2 against the distance between G and GGG nucleobases is reported. Computations predict that, for shorter oligonucleotides, n=1–3 the charge transfer goes intrastrand; the computed hole transfer times being in excellent agreement with their experimental counterpart: simulations yields a straight line with a slope β=0.63 Å as the distance parameter of the Marcus–Levich–Jortner equation [146], to be compared with the experimental value β≈0.6 Å [45].

Interstrand HT becomes about 1 order of magnitude faster than intrastrand HT for n=4. For n≥4, our computations predict that HT rates are almost distance-independent, in good agreement with experimental results. In all cases, hole transfer is predicted to occur via superexchange, since negligible populations have been found on adenine or thymine bridge. According to our simulations, the different distance dependence of HT rates in the short and long distance regimes results from the balance of two contrasting effects: On the one hand, as the number of bridging adenines increases, the energy barrier becomes comparatively lower due to the formation of delocalized polarons, thus favoring hole tunneling along the strand. On the other hand, upon increasing the length of the adenine bridge, the tunneling distance also increases, thus lessening the efficiency of HT, as illustrated in Figure 12.

Yield ratios $P_{G}/P_{\mathrm{GGG}}$ have then been obtained by numerically solving the set of ordinary differential equations (ODEs) of the kinetic model of Figure 10. The experimental yield ratios are compatible with the adopted kinetics model only if $k_{12}$ is set to values of the order of $10^{10}$ s${}^{-1}$. By taking fixed that value for all the cases analyzed by Giese, thus assuming that the rate of the activation process (Step 1) is independent of the bridge length, computations yield the $P_{\mathrm{GGG}}/P_{G}$ ratios reported in Figure 13. Our predictions are in excellent agreement with experimental results, both in the short and in the long range regimes. In particular, a very weak dependence on *n* is predicted for n=4–7.

The computational approach and the set of parameters used for treating Giese sequences are broadly applicable to other DNA oligomers; preliminary results which will appear in a forthcoming paper show that our treatment correctly predicts the yields of DNA oxidative damage in several oligonucleotides studied by Schuster [19].

## 5. Discussion

Hole transfer in DNA is a complex process in which many chemico-physical factors play a role, among which hole site energies, electronic couplings among nucleobases, solvent relaxation, and backbone reorganization times are the most relevant. Hole site energies in DNA are significantly sequence dependent, but that dependence can be reasonably handled using a tight binding approximation in which only interactions among nearest neighbor nucleobases are considered, on condition that reliable electronic couplings are also used. Here, we show that electrochemical well tailored experiments can provide very reliable values of those quantities. Although nucleobase oxidations are usually irreversible processes and prevent the obtainment of standard redox potentials, since voltammetric peaks do not refer to equilibrium conditions, hole site energies and electronic couplings inferred from electrochemical measurements provide a very robust set of parameters for predicting the yields of oxidative DNA damages of several oligonucleotides.

Electrochemical measurements have also shed light on one among the most debated issues on long range hole transfer in DNA: the establishment of charge delocalized domains. Voltammetric measurements have shown a progressive lowering of the first anodic peak potential as the number of adjacent homo-bases (guanine or adenine) increases in DNA sequences. That result is particularly important for adenine: the formation of an AA step considerably lowers the hole site energy, by ca. 0.3 eV, whereas, for GG step, the effect is lower, amounting to about 0.1 eV. That observation is an unambiguous evidence of the establishment of delocalized hole domains in DNA oligonucleotides.

Delocalized domains play a key role in long range hole transfer in DNA, enabling a distance independent regime in adenine rich oligonucleotides, a peculiar property of DNA which should be more deeply explored in organic electronics.

Our developed kinetic model based on hole site energies and electronic couplings inferred from voltammetric measurements is able to reproduce both the yield ratios of damaged products and the correct time scale of HT for the DNA sequences investigated by Giese. Present simulations only include the vibronic ground state localized at the single G as the initial state. However, that approximation should not constitute a severe drawback, inasmuch as the vibrational frequencies of the normal coordinates of single nucleobases that are effectively coupled to hole motion exceed thermal energy at room temperature. Instead, thermal populations of vibrational states possibly affecting hole transfer rates could arise from interbase oscillation modes. Indeed, the electronic coupling for the hole transport in oxidized DNA is expected to depend to a larger extent on variations of the rise coordinate and to a lesser extent on variations of the twist coordinate, interbase motions originating from low frequencies vibrations of DNA backbone [34,103]. We have averaged such thermal effects through the last term of Equation (Equation 6). Nevertheless, a more systematic treatment could rely on the thermo field dynamics theory, in which temperature effects are already included in the Hilbert space [147].

As a final remark, we note that our approach could be easily extended also to the hole transport in G4-quadruplexes, provided reliable hole energies and electronic couplings are available for such complex systems [148]. Work is in progress along that line.

## Figures and Tables

**Figure 1 molecules-24-04044-f001:**
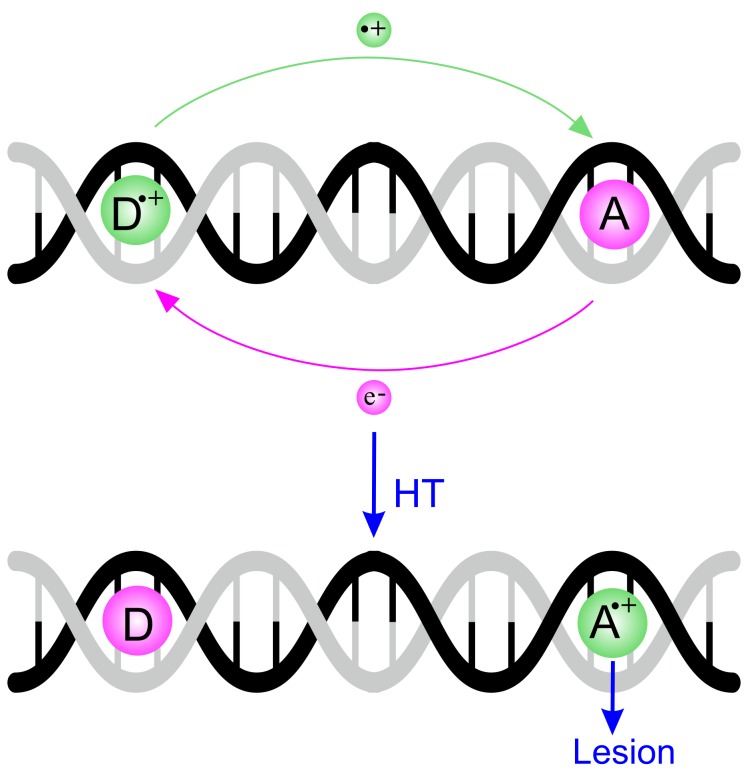
Schematic representation of the hole transport in DNA. The first ionization takes place at the donor site (D), where a hole is generated. The hole migrates through the double helix of DNA, reaching the acceptor site (A), where the final irreversible oxidative damage (lesion) occurs.

**Figure 2 molecules-24-04044-f002:**
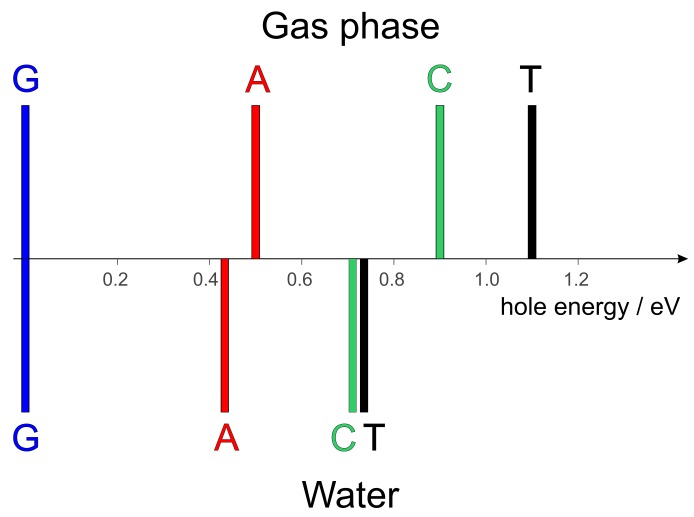
Relative (to guanine) hole energies of DNA nucleobases in the gas phase (top) and in aqueous environment (bottom).

**Figure 3 molecules-24-04044-f003:**
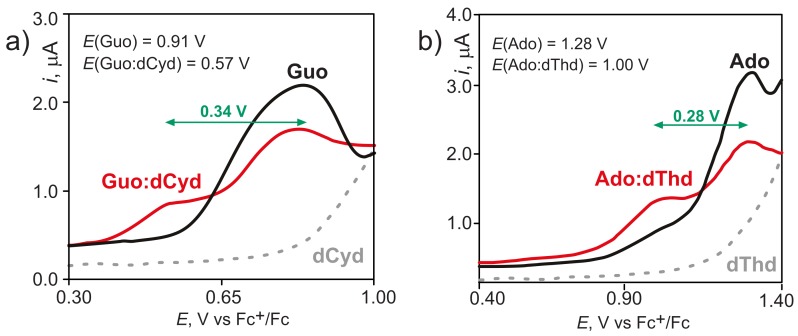
Differential pulse voltammograms of nucleoside derivatives in CHCl${}_{3}$ at 298 K on glassy carbon electrode. (**a**) Solutions containing only Guo 2.0 mM (black line); only dCyd 2.0 mM (dashed gray line); and Guo 2.0 mM and dCyd 2.0 mM (red line). (**b**) Solutions containing only Ado 2.0 mM (black line); only dThd 2.0 mM (dashed gray line); and Ado 2.0 mM and dThd 20.0 mM (red line). Scan rate, 100 mV/s. Supporting electrolyte Bu${}_{4}$NClO${}_{4}$. Oxidation potentials are referred to the Ferrocenium/Ferrocene (Fc${}^{+}$/Fc) redox couple. Green arrows indicate the lowering of the oxidation potential of purine nucleosides upon pairing via H-bond with their complementary pyrimidine nucleosides. Adapted with permission from *J. Am. Chem. Soc.*
**2005**, *127*, 15040–15041 and *J. Am. Chem. Soc.*
**2007**, *129*, 15347–15353. Copyright (2005, 2007) American Chemical Society.

**Figure 4 molecules-24-04044-f004:**
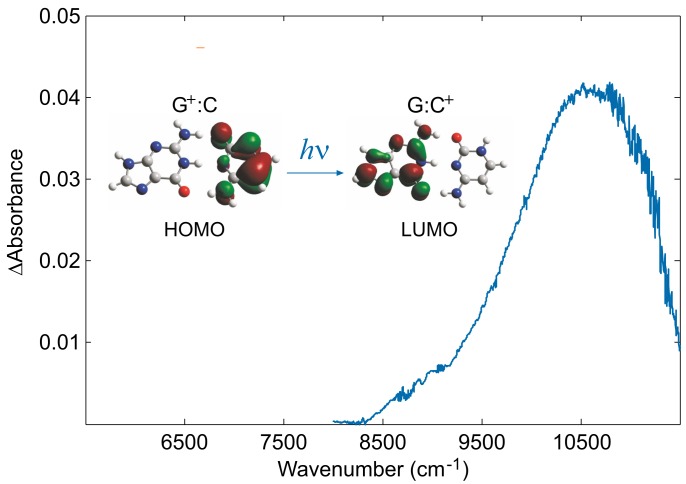
The charge transfer band of the [Guo:dCyd]${}^{+}$ complex recorded in dichloromethane at a controlled potential of +0.57 V vs. Fc${}^{+}$/Fc. It corresponds to the transition from the HOMO, a Kohn–Sham π orbital localized on cytosine, to the LUMO, a $\pi^{*}$ orbital of the guanine moiety. Adapted with permission from *Angew. Chem. Int. Ed.*
**2009**, *48*, 9526–9528. Copyright (2009) Wiley-VCH Verlag.

**Figure 5 molecules-24-04044-f005:**
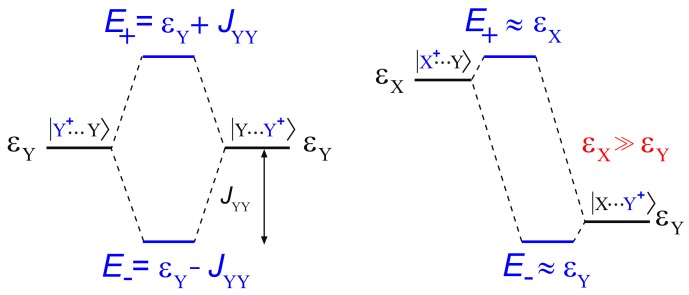
The two limiting cases predicted by the two state model for the ionization energy of two stacked nucleobases. (**Left**) The remotion of an electron from two identical unstacked Y nucleobases pair gives rise to two diabatic states, $|Y^{+}\cdots Y\rangle$ and $|Y\cdots Y^{+}\rangle$ with hole energy $\varepsilon_{Y}$; upon formation of a π stacking interaction, the two states are coupled each other, and the energy levels of the electron hole $\left(E_{+},\;E_{-}\right)$ are shifted up and down with respect to those of unstacked pair by the quantity $J_{\mathrm{YY}}$, representing the strength of the π stacking interaction. (**Right**) If the hole energy of X is far larger than that of Y, then the $J_{\mathrm{XY}}$ coupling term is not effective in lowering the ionization energy of the stack, so that the lowest eigenvalue of H, $E_{-}$, is nearly coincident with $\varepsilon_{Y}$, and the highest eigenvalue of H, $E_{+}$, is nearly coincident with $\varepsilon_{X}$.

**Figure 6 molecules-24-04044-f006:**
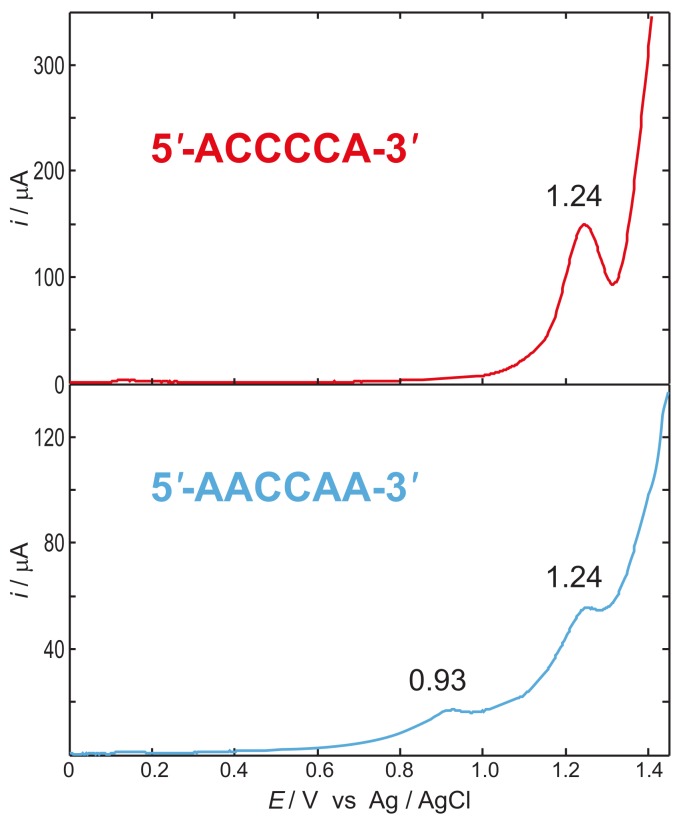
Differential pulse voltammetry of 2.0 mM single stranded 5${}^{\prime }$-ACCCCA-3${}^{\prime }$ (top, red) and 0.50 mM 5${}^{\prime }$-AACCAA-3${}^{\prime }$ (bottom, blue) in 50 mM phosphate buffer solution. Internal reference electrode Ag/AgCl (3.0 M KCl). A lowering of the first anodic signal amounting to 0.31 V is observed in passing from the sequence not containing stacked adenines to the one holding two stacked A’s. Adapted with permission from *J. Phys. Chem. B*
**2013**, *117*, 8947–8953. Copyright (2013) American Chemical Society.

**Figure 7 molecules-24-04044-f007:**
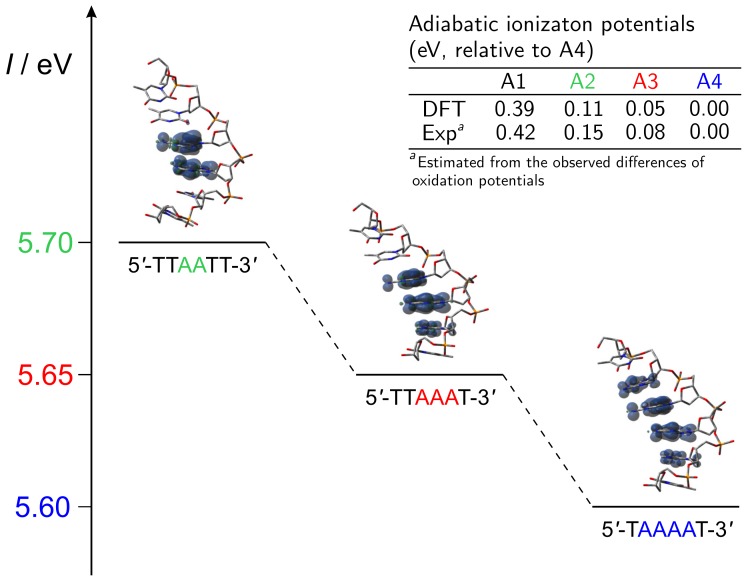
Predicted adiabatic ionization potential and spin densities for the 5${}^{\prime }$-TTAATT-3${}^{\prime }$ (A2), 5${}^{\prime }$-TTAAAT-3${}^{\prime }$ (A3), and 5${}^{\prime }$-TAAAAT-3${}^{\prime }$ (A4) single stranded ionized oligonucleotides. Inset: Comparison between relative computed ionization energies and observed oxidation free energies. The lowering of the oxidation potential observed for sequences lacking G upon increasing the number of stacked adenines has been rationalized in terms of resonance effects: the increasing stability of the hole is due to its delocalization over the entire adenine bridge. Adapted from Ref. [85] with permission from the PCCP Owner Societies.

**Figure 8 molecules-24-04044-f008:**
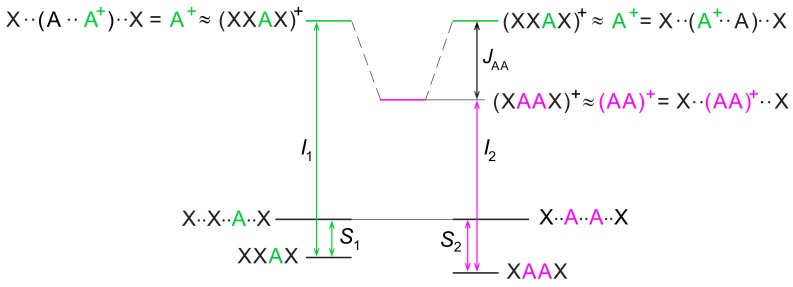
Adenine embedded in the 5${}^{\prime }$-XXAX-3${}^{\prime }$ and 5${}^{\prime }$-XAAX-3${}^{\prime }$ tetrameric single strands, where X is a nucleobase with an ionization energy much larger than that of A. Dotted lines denote non-interacting nucleobases, *S* denote stabilization energies due to stacking interactions in neutral strands and *I* are ionization potentials. Because the ionization energy of X is much larger than that of A, the effect of coupling terms is negligible in (XXAX)${}^{+}$, so that the hole is fully localized on A. Since A in XXAX and AA in XAAX experience almost identical environments, the difference of the stacking energy in XXAX and XAAX neutral strands is expected to be small ($S_{1}-S_{2}\approx 0$); therefore, the difference of hole energies of XXAX and XAAX oligonucleotides is well approximated by the difference of ionization energies, $J_{\mathrm{AA}}\approx I_{1}-I_{2}$. Reproduced from Ref. [62] with permission from the PCCP Owner Societies.

**Figure 9 molecules-24-04044-f009:**
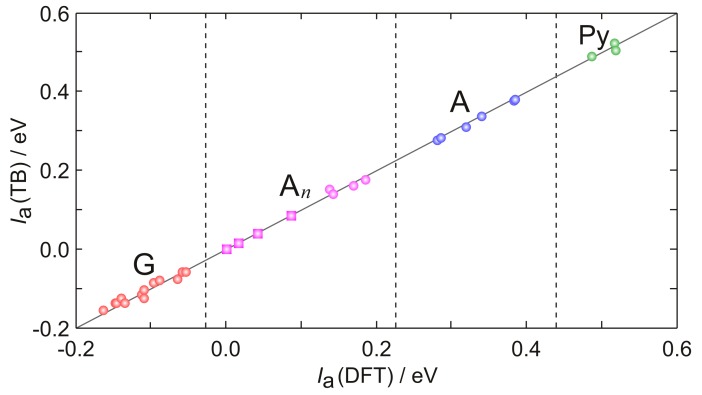
Comparison of adiabatic ionization energies (relative to XXGX, X being 6-azauracil) predicted by DFT (abscissa) and TB (ordinate) computations for tetrameric single stranded DNA oligonucleotides (circles). Hole energies for TA${}_{n}$T (n=3–6) sequences (squares) have been computed only at the TB level. Full line has null intercept and unitary slope. Dashed lines separate the regions of potential corresponding to the oxidation of G (red), A${}_{n}$ tracts (violet), A (blue) and pyrimidines (green). Very similar oxidation patterns were found by differential pulse voltammograms recorded in buffered aqueous solutions for single and double stranded DNA oligonucleotides and DNA itself (see Refs. [47,59,124,125]). Reproduced from Ref. [62] with permission from the PCCP Owner Societies.

**Figure 10 molecules-24-04044-f010:**
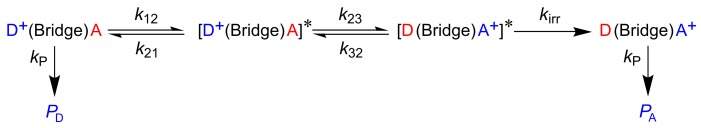
The kinetic scheme adopted to model HT in DNA. D${}^{+}$(Bridge)A and D(Bridge)A${}^{+}$ indicate the initial and the finale state, possibly giving rise to oxidation product $P_{D}$ and $P_{A}$. [D${}^{+}$(Bridge)A]* and [D(Bridge)A${}^{+}$]* denote the ensembles of structures in which the hole donor and acceptor are in vibronic resonance with each other, so that $k_{23}=k_{32}$. $k_{21}=k_{\mathrm{irr}}$ and $k_{P}$ have been taken from experimental data; $k_{23}$ has been computed by resolving the time dependent Schrödinger equation; $k_{12}$ is an adjustable parameter to be inferred by comparing computed and experimental product yield ratios.

**Figure 11 molecules-24-04044-f011:**
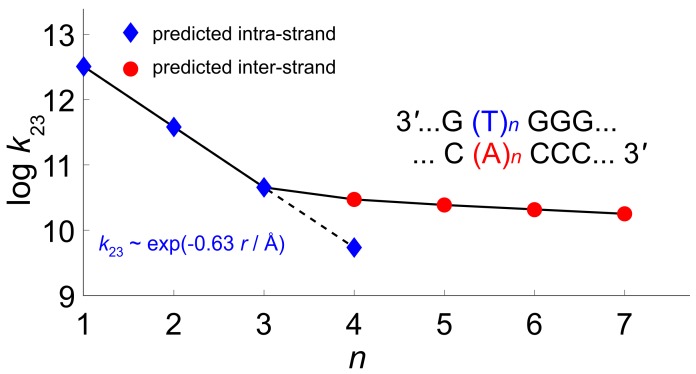
Predicted rate constants $k_{23}$ for the HT step in 3${}^{\prime }$-G(T)${}_{n}$GGG-5${}^{\prime }$ DNA sequences, as a function of the number of bases separating the donor (G) and the acceptor (GGG) sites; blue diamonds, Intrastrand; red circles, Interstrand HT via the (A)${}_{n}$ bridge of the complementary helix. The predicted parameter of the Marcus–Levich–Jortner equation (in blue) for the short distance regime (n=1–3) is in excellent agreement with its experimental counterpart. For n≥4, HT is predicted to occur interstrand, mediated by the adenine bridge. Adapted with permission from *J. Phys. Chem. Lett.*
**2019**, *10*, 1845–1851. Copyright (2019) American Chemical Society.

**Figure 12 molecules-24-04044-f012:**
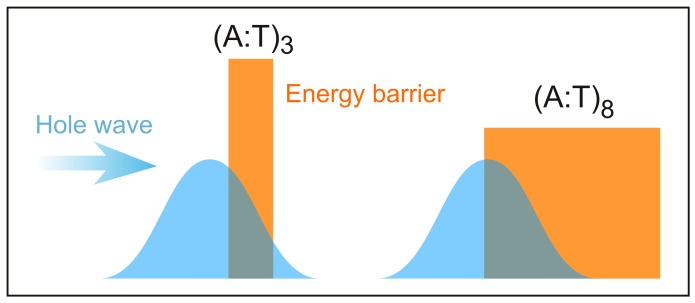
The contrasting effects of the bridge in the HT for the sequences studied by Giese: In short sequences (**left**), superexchange is favored by the short tunneling distance and disfavored by high barriers. In longer sequences (**right**), barrier height decreases due to formation of delocalized $A_{n}^{•+}$ polarons thus favoring the HT, but tunneling distance increases, thus attenuating the efficiency of HT at the same time.

**Figure 13 molecules-24-04044-f013:**
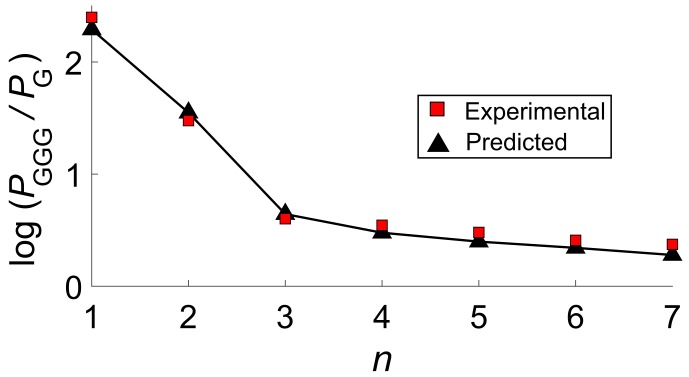
Predicted (black triangles) and experimental (red squares) $P_{\mathrm{GGG}}/P_{G}$ yield ratios for the HT in *d*-G(T)${}_{n}$GGG as a function of the number of bases separating the donor and the acceptor G sites. Computed values have been obtained by setting $k_{12}=10^{10}$ s${}^{-1}$ for all the investigated sequences, obtained by solving the ODEs equations for the kinetic scheme of Figure 10. Adapted with permission from *J. Phys. Chem. Lett.*
**2019**, *10*, 1845–1851. Copyright (2019) American Chemical Society.

**Table 1 molecules-24-04044-t001:** Oxidation free energies ($\Delta G^{\mathrm{ox}}$) and adiabatic ionization energies (*I*) of DNA constituents relative to guanine or its derivatives. N, nucleobases; Ns, nucleosides; Nt, nucleotides. All data are expressed in eV.

	*I*(N)${}^{a}$	$\Delta G^{\mathrm{ox}}$(Ns)${}^{b}$	$\Delta G^{\mathrm{ox}}$(N)${}^{c}$	$\Delta G^{\mathrm{ox}}$(Nt)${}^{c}$	$\Delta G^{\mathrm{ox}}$(Ns)${}^{d}$	$\Delta G^{\mathrm{ox}}$(Ns) ^e^	$I_{g}$(N)${}^{f}$
A	+0.41	+0.5	+0.27	+0.30	+0.47	+0.15	+0.49
C	+0.74	+0.8	+0.61	+0.57	+0.65	+0.38	+0.91
T	+0.75	+0.8	+0.45	+0.52	+0.62	+0.31	+1.10

^*a*^ Ref. [62], density functional theory (DFT) computations including aqueous environment, via‘the polarizable continuum model (PCM) [63]. ^*b*^ Ref. [64], PES measurements in water integrated with ab initio computations. ^*c*^ Ref. [17], voltammetry, water pH 7. ^*d*^ Ref. [58], voltammetry, acetonitrile. ^*e*^ Ref. [65], nanosecond spectroscopy, data adjusted as in Ref. [64]. ^*f*^ Ref. [66], gas phase photoionization mass-spectrometry.

**Table 2 molecules-24-04044-t002:** Hole site energies ($\varepsilon_{Y}$, eV, relative to G${}^{+}$) and electronic coupling parameters for stacked 5${}^{\prime }$-YZ-3${}^{\prime }$ base pairs ($J_{\mathrm{YZ}}$, eV). Y and Z denote native DNA nucleobases. All values are taken from Ref. [62].

Y	$\varepsilon_{Y}$	$J_{\mathrm{YG}}$	$J_{\mathrm{YA}}$	$J_{\mathrm{YC}}$	$J_{\mathrm{YT}}$
G	0.00	0.09	0.15	0.23	0.14
A	0.43	0.15	0.24	0.16	0.08
C	0.68	0.23	0.16	0.12	0.12
T	0.70	0.14	0.08	0.12	0.12

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
