# Peer review of "The Dynamics of Hole Transfer in DNA"

_molecules, 2019, doi:10.3390/molecules24224044_

Round 1
Reviewer 1 Report
The manuscript reports an interesting review dealing with the authors works on the determination of hole mobility in DNA and their attempts, in many cases quite succesfull, to provide a comprehensive description of this important phenomenon. Asd such I think this review is a welcoming addition and can be extremely useful for the community.
I only have some minor considerations that I urge the authors to take into account before considering the manuscript for final pubblication.
-The introduction starts quite abruptly I think a paragraph reporting analysis of the structure of DNA, its coupling and its biological role would improve the presentation. Besides it should be important to clarify as much as possible when the authors refer to canonical double-helix DNA or to other non-canonical forms, single stranded, G4, etc....
-At line 57 the authors explicitly mention solvation effects and hydrogen bonds. I think it would be appropriate to mention here also the couplign with other nucleobases due to pi-stacking and dispersion that will constitute the main subject of the following section.
-The calculations of the solvation effects on the DNA oxidation potential and their comparison with experiment is welcomed and well done. However, the authors could mention the fact the in double strand DNA the nucleopbases are embedded in a strong hydropobic environment as opposed to the charged backbone.
-Line 156 I believe the sentence "observed of adenine" should read "observed for adenine"
-Once again the results of the modeling and simulations are very promising. However, it would be nice to enlarge the discussion to assess, at least from the authors point of view, the effects of dynamical deformation (that in DNA can be also of large scale) and more generally vibrational degrees of freedom on the values of couplings and site energy, and hence ultimately on hole mobility. This could also be interesting to be assessed for their kinetic model.
-I believe that some more details concerning the quantum dynamic protocol used should be provided to allow a full and consistent picture and carefully judging of the relevance of the model and its approximations.
-Is it possible that the observed switching in the transfer regime when increasing the strand lenght could be due to the increased flexibility of the oligomer and else the breaking of coherent regions due to thermal motion? A comment would be welcomed.
-The discussion is globally well performed. However, I would suggest to stress out the role of molecular modeling and simulation in solving such complex problems.
Reviewer 2 Report
Peluso et al., based on their own studies, gave a summary about hole transportation in DNA. Factors that influence hole transportation such as hole energies of the nucleotides, distances, and sequences are discussed. The authors also discussed computational simulation about hole transporting. In general, the manuscript is very-well written and I suggest to publish after minor changes.
1, For short DNA oligonucleotides in aqueous solution, if this is the case, how different solvent (buffer, pH) and ionic strength (different ions and different concentration, e.g., Na+, K+, Mg2+) may influence the HT?
2, In the abstract, the authors mentioned the study “is important not only for its biological consequences, but also for assessing the properties of DNA in redox sensing or labeling”, the current manuscript which is based on their work in the last decade may not cover topics of application, it may be still necessary to give a brief outlook in the section 5 about research challenges and potential applications (e.g., in DNA damages or even DNA spins lattices?).
